# The Reliability of Ultrasonographic Assessment of Depth of Invasion: A Systematic Review with Meta-Analysis

**DOI:** 10.3390/diagnostics13172833

**Published:** 2023-09-01

**Authors:** Marco Nisi, Stefano Gennai, Filippo Graziani, Rossana Izzetti

**Affiliations:** Department of Surgical, Medical and Molecular Pathology and Critical Care Medicine, University of Pisa, 56123 Pisa, Italy

**Keywords:** oral neoplasm, ultrasonography, oral squamous cell carcinoma, intraoral ultrasound, systematic review, meta-analysis

## Abstract

Depth of invasion (DOI) has been recognized to be a strong prognosticator for oral squamous cell carcinoma (OSCC). Several diagnostic techniques can be employed for DOI assessment, however intraoral ultrasonography has been increasingly applied for the intraoral evaluation of OSCCs. The aim of the present study is to review the evidence on the application of intraoral ultrasonography to the assessment of DOI in patients affected by OSCC. A systematic electronic and manual literature search was performed, and data from eligible studies were reviewed, selected, and extracted. The studies had to report the correlation between DOI estimated with ultrasonography versus histopathology. A meta-analysis was conducted on the quantitative data available. Sixteen articles were included in the review following the screening of the initial 228 studies retrieved from the literature. The meta-analysis showed a significant correlation between ultrasonographic and histopathologic measurements (*p* < 0.01). The studies were all at low/moderate risk of bias. Ultrasonography appears a valuable tool for DOI assessment.

## 1. Introduction

The parameter of depth of invasion (DOI), defined as the distance between the normal mucosal surface and the deepest margin of a neoplastic lesion in the tissues, has been proven to be a valid prognosticator of oral squamous cell carcinoma, and is recognized as a T-stage modifier by the eighth edition of the American Joint Committee on Cancer (AJCC) criteria [1,2]. The assessment of DOI can be predictive of cervical lymph nodes involvement, as well as facilitating the achievement of clear surgical margins, allowing an improved local disease control and preventing recurrence [3].

DOI can be assessed either on diagnostic imaging datasets prior to tumor excision or on histopathological samples following surgical resection [4]. While pathologic DOI is considered the reference standard for tumor depth assessment, previous evidence has highlighted how magnetic resonance and ultrasonography perform extremely well in preoperative DOI assessment, showing high correspondence with histology [5,6]. Importantly, intraoral ultrasonography has been reported to have the highest correlation with pathological DOI compared to other diagnostic techniques [7].

The aim of the present systematic review is to analyze the evidence behind the application of intraoral ultrasonography to the assessment of DOI in patients affected by oral squamous cell carcinoma.

## 2. Materials and Methods

### 2.1. Protocol Development and Eligibility Criteria

The protocol for the present study was prepared according to the Preferred Reporting Items Systematic review and Meta-Analyses (PRISMA) [8,9,10] and registered in PROSPERO (CRD42023446434). The following focused question was phrased:

“What is the reliability of ultrasonography in the assessment of depth of invasion of oral squamous cell carcinoma?”

Articles to be included had to follow the following PICO:

(P) Type of participants: patients with a diagnosis of oral squamous cell carcinoma eligible for surgical treatment;

(I) Type of interventions: assessment of depth of invasion with intraoral ultrasonography;

(C) Comparison between interventions: depth of invasion measurement on histology;

(O) Type of outcome measures: correlation between ultrasonographic DOI and pathologic DOI.

Systematic reviews and review articles were not included. No time limitations were applied. Only articles in English were included.

### 2.2. Literature Search

The electronic search was applied to the Cochrane Oral Health Group specialist trials, MEDLINE via PubMed, and EMBASE (SG) up to June 2023. A combination of MeSH terms and free text words was employed:

((“Mouth Neoplasm”[Mesh] OR “Oral Neoplasm” OR “Oral squamous cell carcinoma” OR “Oral Carcinoma” OR “Oral Cancer”) AND (“Ultrasonography”[Mesh] OR “Ultrasound” OR “Intraoral ultrasonography” OR “Intraoral Ultrasound”) AND (“Neoplasm Invasiveness”[Mesh] OR “Depth of Invasion”))

Trials databases such as clinicaltrials.gov were searched. The bibliographies of review articles and relevant papers were checked (RI, MN).

### 2.3. Study Selection and Data Collection

Eligibility assessment was performed through title and abstract analysis of the search results, with an initial screening performed by two reviewers (RI, MN) for possible inclusion in the review. The two reviewers were calibrated for study screening against a third reviewer expert in systematic reviews (SG). Calibration consisted in the independent validity assessment of 20 titles and abstracts retrieved from the search until a κ-score > 0.8 was achieved. The articles selected through title and abstract analysis were then assessed through full text analysis. Unclear abstracts were included in the full text analysis to avoid the exclusion of potentially relevant articles. Title and abstract analysis was performed in June 2023.

Inclusion criteria for the title and abstract analysis were the following:Patients with a diagnosis of OSCC and eligible for surgery;Patients evaluated with intraoral ultrasound for DOI assessment;Studies reporting histological DOI evaluation and correlation with ultrasonographic DOI;Manuscripts published in English.

Exclusion criteria for the title and abstract analysis were the following:Subjects with conditions other than OSCC;Patients not evaluated with ultrasound;Assessment of imaging parameters other than DOI;Lack of reporting of histopathologic DOI;Descriptive studies not reporting the correlation between ultrasonographic and pathologic DOI;Studies that could not be classified as case–control studies, cohort studies, cross-sectional studies, case-series trials, controlled trials, or randomized controlled trials.

Full texts of the selected articles were then retrieved and independently assessed by two reviewers against the stated inclusion criteria (RI, MN). The articles had to follow the inclusion criteria to be included in the systematic review. The same exclusion criteria were employed for the full text analysis, together with absence of reporting of any of the studied outcomes. In cases of disagreement, the full text was discussed with a third experienced reviewer (SG). Data of the included articles were extracted and collected through an ad hoc extraction sheet (RI, MN). Full text inclusion was performed in June 2023 and full text data extraction by mid-July 2023. The reviewers conducted all quality assessments independently.

### 2.4. Risk of Bias in the Included Studies and Quality Assessment

The quality assessment and the risk of bias of the included studies was performed following the criteria of the ROBINS-I tool (Risk Of Bias In Non-randomized Studies—of Interventions) evaluating selection, comparability, and outcome domains for each study [11]. In cases of critical or serious judgment, the study was considered at high risk of bias.

### 2.5. Summary Measures and Synthesis of the Results

Data synthesis was presented through evidence tables addressing study characteristics and main conclusions. The performance of possible meta-analysis was decided on the basis of the similarity and availability of quantitative data. Results were expressed as weighted mean difference (WMD) and 95% confidence interval (CI) for continuous outcomes using both random and fixed models.

The meta-analysis was performed with the Fisher r-to-z transformed correlation coefficient as the outcome measure. Heterogeneity was assessed via Q-test, I^2^, and tau^2^, the latter assessed through the restricted maximum likelihood estimator [12]. If tau^2^ > 0 was detected, a prediction interval for the true outcomes was also provided. The evaluation of potential outliers and/or influential studies in the context of the model was performed with the studentized residuals and Cook’s distances. If a studentized residual larger than the 100 × (1 − 0.05/(2 × k))^th^ percentile of a standard normal distribution was found, the study was considered a potential outlier. If a Cook’s distance larger than the median plus six times the interquartile range of the Cook’s distances was found, the study was considered influential. Funnel plot asymmetry was checked through rank correlation and the regression tests. OpenMeta [Analyst] (http://www.cebm.brown.edu/open_meta/open_meta/open_meta, accessed on 1 June 2023) or other equivalent software for meta-analysis were employed, and the results were graphically illustrated and summarized with forest plots.

## 3. Results

### 3.1. Study Selection

The electronic search retrieved a total of 228 articles (205 articles from the electronic database search and 23 articles from the hand search) published up to June 2023. After the removal of duplicates, title and abstract analysis was performed on 211 articles. One hundred and seventy-eight articles were excluded following the screening of titles and abstracts. Full text analysis was performed on the remaining 33 articles, and 17 articles were further excluded. The final review included 16 articles [13,14,15,16,17,18,19,20,21,22,23,24,25,26,27,28], which all met the criteria for inclusion in the meta-analysis (Figure 1).

### 3.2. Population and Studies Characteristics

The study population consisted of 729 patients, with a mean age of 61.82 ± 5.56 years. Information on gender distribution was available for 14 out of 16 articles [13,15,16,17,18,19,20,21,22,23,24,25,26,28], accounting for a population of 696 patients, 420 males and 276 females.

In 12 studies [13,14,15,17,19,20,22,23,24,25,26,28], the AJCC/UICC classification was employed for tumor staging, while two studies did not report the classification system employed.

In 13 studies [13,15,16,17,20,21,22,23,24,25,26,27,28], ultrasonography was performed preoperatively. The frequencies employed ranged between a minimum of 5 and a maximum of 70 MHz (Table 1).

### 3.3. Synthesis of the Main Findings of the Included Studies

Kurokawa et al. [13] included 28 patients with OSCC of the tongue. The authors reported a correlation between DOI assessed with 7.5 MHz ultrasonography and T stage, tumor size, N stage, type of invasion, muscular invasion and deep invasive front grading. No correlation was found with growth type, differentiation, and Anneroth’s malignancy. Ultrasound measurements correlated with histology. Other diagnostic techniques (computed tomography and magnetic resonance) tended to overestimate tumor dimensions.

Songra et al. [14] employed ultrasonography (frequency range 5–10 MHz) to assess deep margins intraoperatively half way through surgical resection. The authors reported good agreement between ultrasound and histology when applying a 5 mm threshold to indicate clear surgical margins. The technique was reported to have 83% sensitivity and 63% specificity.

In the study by Mark Taylor et al. [15], intraoral ultrasound (10–12 MHz) was performed in patients with biopsy proven squamous cell carcinoma of the tongue or floor of the mouth. The authors found that in cases of DOI < 5 mm, none of the patients presented positive lymph nodes, while in the presence of DOI ≥ 5 mm, 65% of the patients had nodal metastases. The authors concluded that preoperative ultrasonography was accurate in the assessment of tumor dimensions, and that in the presence of DOI ≥ 5 mm elective neck dissection is recommended.

Iida et al. [16] compared preoperative ultrasound measurement of DOI performed with a 16 MHz probe with histological DOI. The authors discriminated the accuracy of the technique depending on tumor size. In cases of superficial tumors, the comparison between ultrasonography and histology was 1 mm in 64.1% of cases and 2 mm in 92.3% of cases. In the presence of in situ OSCCs, the DOI ranged between 0.8 mm and 1.6 mm. Ultrasonography appeared reliable in tumors with DOI ≤ 5 mm, corresponding to T1 clinical staging according to the eighth edition of the AJCC, and was comparable to histology when analyzing superficial tongue carcinomas. The authors concluded that intraoral ultrasonography may constitute a diagnostic supplement especially in cases of superficial tumors where discordance between radiographic-derived and clinically derived values is encountered.

Noorlag et al. [17] analyzed a retrospective cohort of 209 patients with T1-T2 OSCC of the tongue, evaluated preoperatively with intraoral ultrasonography (15 MHz) and magnetic resonance (1.5–3.0 T). Ultrasonography showed a mean absolute difference with histology of 1.6 mm in smaller tumors and of 4.7 mm in larger tumors. Magnetic resonance showed a mean absolute difference with histology of 3.2 mm. The authors encountered an overall underestimation employing both diagnostic techniques compared to histological DOI. Among the presumable reasons for such a discrepancy, the authors listed (i) the timespan between imaging and surgery, which in some cases was more than four weeks; (ii) the pressure applied to the tumor during ultrasound scan; and (iii) tumor shrinkage following formalin fixation and/or slicing errors during specimen processing for histology. The conclusions reported a good correlation between magnetic resonance and intraoral ultrasonography measurements with histology. Ultrasonography showed higher accuracy in tumors with pathological DOI ≤ 10 mm (T1–T2 according to the eighth edition of the AJCC criteria) compared to magnetic resonance which tended to overestimate DOI, while in tumors >10 mm (T3) the accuracy of intraoral ultrasonography decreased.

Yoon et al. [18] performed tumor resection of OSCCs of the tongue under ultrasound guidance (7–15 MHz frequency) in 20 patients. Mean ultrasonographic DOI was 6.6 mm ± 3.4 mm and histopathologic DOI was 6.4 mm ± 4.4 mm, with a high correlation between the two measurements. Intraoperative application of ultrasonography resulted in an improvement in the achievement of clear resection margins and in the performance of elective neck dissection in the presence of DOI > 4 mm.

Bulbul et al. [19] included 23 patients with T1-T3 OSCC of the tongue and performed ultrasound (7–15 MHz)-guided tumor resection, and compared the surgical outcomes to a control group composed by 21 patients with T1-T3 OSCC surgically treated without ultrasound guidance. The mean closest margins for the ultrasound group were 6.3 mm ± 2.8 mm and 4.3 mm ± 2.7 mm for the control group. The mean deep margins for the ultrasound group were 8.5 mm ± 4.9 mm and 6.7 mm ± 3.8 mm for control group. Ultrasonographic guidance allowed for obtaining improved overall and deep margin clearance, 78% negative (≥5 mm) deep margins, and the absence of frankly positive deep margins.

Filauro et al. [20] performed a retrospective evaluation of 49 patients with T1-T3 OSCC who underwent intraoral ultrasound scan (7–15 MHz) and/or magnetic resonance imaging (1.5–3.0 T). The mean value of DOI was 7.0 mm with ultrasonography, 7.2 mm with magnetic resonance, and 7.3 mm with histology. Magnetic resonance provided a correct staging in 64% of cases, while intraoral ultrasound correctly staged all patients. For elective neck dissection, indicated in the presence of pathological DOI ≥ 4 mm, a 100% sensitivity was found for both the techniques, while 73% specificity for magnetic resonance and 47% specificity for ultrasonography were detected. The best cut-off for elective neck dissection was a radiological DOI ≥ 5 mm. Overall sensitivity was 92% for magnetic resonance and 87% for ultrasonography, while specificity was 93% for magnetic resonance and 76% for ultrasonography. While recognizing a good performance of ultrasonography compared to magnetic resonance, the authors recognized as limits to ultrasonography applicability the operator-dependency and the limited imaging capability for lesions close to bony structures or in the posterior half of the oral cavity.

Harada et al. [21] compared clinical DOI and radiological DOI assessed with ultrasound, MRI before biopsy, and MRI after biopsy. The authors performed a correction on pathological DOI values as a 10.3% shrinkage of the specimen after preparation was estimated. MRI before biopsy showed the highest concordance with clinical DOI, with a slight overestimation. Ultrasonography tended to underestimate clinical DOI. MRI after biopsy showed an overestimation of clinical DOI related to the inflammatory reaction of tongue muscles following bioptic sampling.

Izzetti et al. [22] assessed the correlation between ultrasonography performed at 70 MHz frequency and histology in a pilot sample of 10 patients affected by OSCC. A significant correlation was found between the two techniques. A 0.14 mm overestimation was registered for DOI values assessed through ultrasonography.

Rocchetti et al. [23] performed an ultrasonographic assessment of OSCC of the oral cavity using 8–17 MHz frequencies in 32 patients. The authors reported the following values for ultrasound assessment of tumor depth: 93.1% sensitivity, 100% specificity, 100% PPV, and 60% NPV. According to their results, ultrasonography appeared effective especially in the assessment of early-stage tumors.

Caprioli et al. [24] compared preoperative DOI assessed through MRI and ultrasonography (8–22 MHz frequency) with pathological DOI in 41 patients with tongue OSCC. While magnetic resonance tended to overestimate, ultrasonography showed a 92.31% sensitivity and 82.14% specificity in predicting a pathological DOI ≥ 4 mm, with a 100% specificity and a 94.7% sensitivity in discriminating an invasive cancer.

Nilsson et al. [25] enrolled 40 patients with biopsy-proven primary T1-T3 OSCC of the tongue and floor of the mouth and performed preoperative ultrasound employing an 18 MHz equipment. The authors compared DOI measurements obtained with ultrasound, magnetic resonance, computed tomography and histology. A DOI assessment was performed in all the patients employing ultrasonography, in 79% of patients with magnetic resonance and in 5% of patients with computed tomography. For magnetic resonance, motion artifacts and reduced tumor dimensions hindered DOI evaluation, while computed tomography was prone to artifacts. A comparison with histology revealed an error of 0.5 mm for ultrasonographic measurements, which further decreased to 0.1 mm in cases of T1-T2 tumors. A mean overestimation of 3.9 mm was reported for magnetic resonance, which appeared more reliable when assessing T3 tumors.

Takamura et al. [26] evaluated 48 patients with T1-T2 tongue OSCC (T1N0: 26 patients; T1N1: 17 patients, T2N0: 2 patients, and T2N1: 3 patients) and compared DOI as assessed with computed tomography, magnetic resonance (1.5 T), and ultrasonography (7–13 MHz). Computed tomography showed a mean difference of 2.7 mm between the histopathological DOI and radiological DOI, while for magnetic resonance the mean difference was around 2 mm. Ultrasonography DOI measurement differed by a mean of 0.2 mm, being the most accurate diagnostic imaging measurement method.

Au et al. [27] performed intraoral ultrasound assessment of biopsy-confirmed OSCC of the tongue in clinically nodal-negative patients treated with resection. In total, 19 patients were assessed with intraoral ultrasonography, and a strong correlation between ultrasonography and histology was found (*p* < 0.001), with a 90% sensitivity and 78% specificity.

Kumar et al. [28] performed intraoral ultrasonography (6–13 MHz) and contrast-enhanced magnetic resonance (1.5 T) in patients affected by T1-T3 biopsy-proven tongue OSCC. Ultrasonography was superior to magnetic resonance in T1 tumors with pathological DOI ≤ 5 mm compared to T2 tumors (DOI 5–10 mm).

### 3.4. Meta-Analysis

All 16 studies resulting from full text analysis were eligible for inclusion in the meta-analysis. The Fisher r-to-z transformed correlation coefficients range was 0.6931–2.3235, and all the estimates were positive. Based on the random-effects model, µ was 1.4041 (95% CI: 1.1783 to 1.6300), with the average outcome significantly differing from zero (z = 12.1877, *p* < 0.0001). The correlation coefficients appeared heterogeneous according to the Q-test (Q(15) = 87.3755, *p* < 0.0001, tau^2^ = 0.1740, I^2^ = 87.5430%), with a 95% prediction interval between 0.5560 and 2.2523. However, the correlation coefficients of the included studies were in the same direction as the estimated average outcome. None of the studies had a value larger than ± 2.9552 after studentized residuals analysis, thus revealing the absence of outliers. According to the Cook’s distances, none of the studies could be considered to be overly influential. The regression test indicated funnel plot asymmetry (*p* = 0.0415) but not the rank correlation test (*p* = 0.0517). (Figure 2).

### 3.5. Risk of Bias Assessment

All 16 studies showed a moderate/low risk of bias. Four studies showed a high risk of bias in the selection of participants, as the classification criteria employed for OSCC diagnosis were not reported (Figure 3).

## 4. Discussion

The present results support the application of ultrasonography as a reliable tool in the assessment of DOI. Although the body of evidence is limited and the available literature is inhomogeneous in terms of frequencies and protocols employed, an overall consistency in the reporting is observed, suggesting a correlation between ultrasonographic DOI assessment and histology.

The role of preoperative DOI assessment has been extensively investigated in the literature, and its validation as a prognosticator for OSCC is represented by the inclusion in the eighth edition of the AJCC staging as a T-category modifier. Importantly, DOI appears extremely valuable in the presence of early OSCC, improving prognostication. Murthy et al. reported that increasing DOI is associated with poorer prognosis in T1-T2 OSCC, although a plateau in estimated survival rates is observed for tumor DOI > 5 mm [30]. Moreover, 5 mm DOI was reported to be a cut-off for the presence of occult nodal metastases in T2 patients [30]. From this perspective, DOI is an independent predictor of nodal metastases, and its inclusion in the staging system improved the decision-making of elective neck dissection, especially in patients with early OSCC [31].

The application of the eighth AJCC has been reported to result in an upstaging of patients with early OSCC compared to the seventh edition, improving the discrimination among pT1, pT2, and pT3 for disease-free survival and five-year overall survival [31]. However, Tsai et al. described a more favorable prognosis for pT1N1 than pN2-3N1 in stage III OSCC, and highlighted the need for a re-classification and a down-staging patients with pT1N1 disease [32]. Similarly, Kang et al. suggested a downstaging of pT4bN0-2 and pT1-2N3b to pStage IVA due to their less adverse prognosis [33]. Berdugo et al. suggested the incorporation of tumor size along with DOI for pT staging, describing tumor dimensions as a robust prognosticator limitedly dependent on histological variables [34]. Conversely, Newman et al. proposed a distinction between two prognostic groups in the pT3N0M0 stage depending on the DOI, with treatment escalation for deeper tumors [35]. As deeper DOI is a predictor of poorer relapse-free and overall survival, it has also been hypothesized to subdivide stage III OSCC based on DOI cut-off [36]. Undoubtedly, the eighth edition of the TNM clinical staging system has improved the ability to discriminate and prognosticate OSCC, by identifying patients with higher mortality rated through the application of clinical DOI and extranodal extension [2].

Ultrasonography has seen an increasing application in several medical fields, due to its ability to provide diagnostic information without the application of ionizing radiations, at a relatively lower cost compared to other diagnostic techniques [37,38,39,40,41,42]. Ultrasonography has been reported to be extremely high-performing in the preoperative assessment of OSCC, as it is estimated to have a 91–93% sensitivity [43]. In particular, ultrasonography finds indication in the presence of small tumors, which are not detectable through other diagnostic imaging techniques such as computed tomography and magnetic resonance [44]. The results of the present study confirm the good performance of ultrasonography in assessing DOI, and the available body of literature supports its application for the preoperative and/or intraoperative evaluation of OSCC.

The present study has some limitations. First, the variability of frequencies employed and ultrasound acquisition protocols may hinder comparison between studies, and thus the drawing of firm conclusions. However, it could be observed that the current literature consistently reports the use of linear probes, although the variability in the frequencies employed may hinder the recommendation of a specific ultrasound frequency. Importantly, although some differences were detected in terms of the timing of ultrasonographic scan, it could be hypothesized that preoperative ultrasonography may prove beneficial for surgical planning. Nevertheless, the evidence on the intraoperative ultrasound acquisition supports a role of this technique in tumor resection with clear margins. Secondly, although the definition for DOI was cautiously screened in order to discriminate studies reporting on other parameters (e.g., tumor thickness), great variability is encountered in the literature, hindering further assumptions regarding the measurement of DOI depending on the ultrasound frequencies employed. Finally, some of the included studies reported the application of ultrasonography to sites other than tongue, thus potentially representing a confounding factor. Nevertheless, our results support a role for intraoral ultrasonography in evaluating DOI.

## 5. Conclusions

Ultrasonographic assessment of DOI is a reliable tool in the evaluation of OSCC and the studies present in the literature consistently report high correlation coefficients with histopathology. Further studies aimed at improving the definition of acquisition protocols and the frequencies to be used are needed.

## Figures and Tables

**Figure 1 diagnostics-13-02833-f001:**
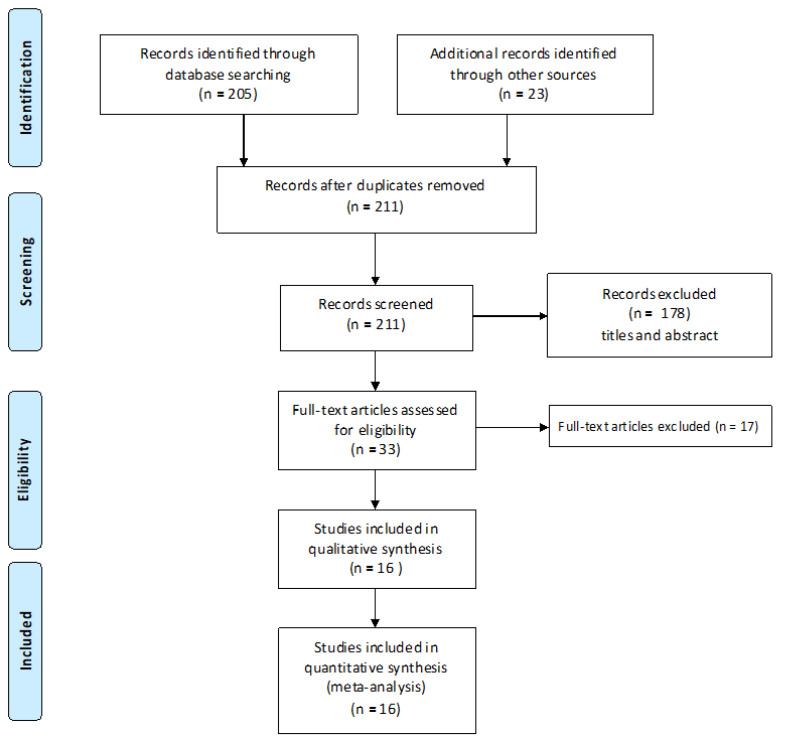
Study flowchart.

**Figure 2 diagnostics-13-02833-f002:**
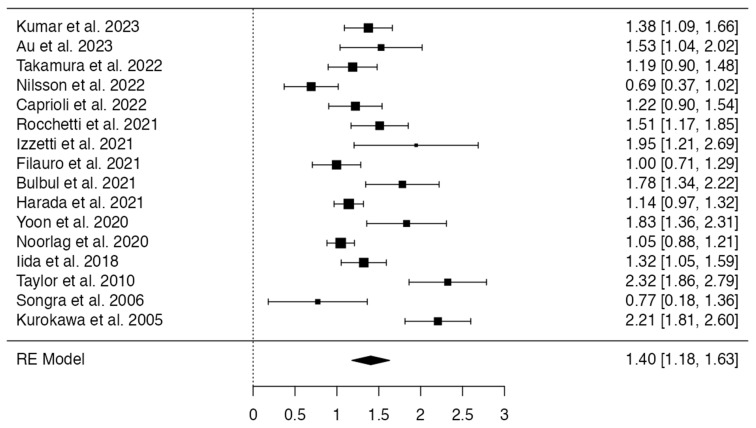
Forest plot of the meta-analysis.

**Figure 3 diagnostics-13-02833-f003:**
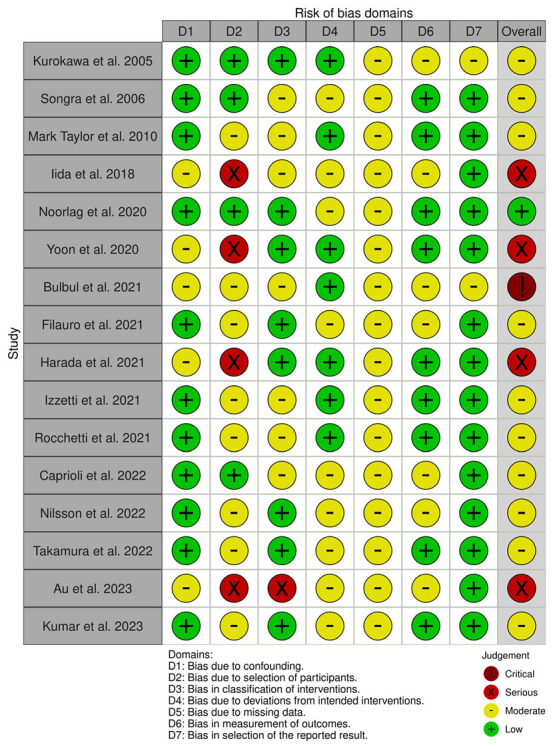
Risk of bias assessment of the 16 articles included in the review [13,14,15,16,17,18,19,20,21,22,23,24,25,26,27,28]. Table generated with *robvis* tool [29].

**Table 1 diagnostics-13-02833-t001:** Evidence table of the included studies.

Authors	Design	Patients	M:F	Mean Age	Staging Method	Tumor Location	T Stage (Patients No.)	Timing	Us Equipment	US Frequency	Correlation P-DOI/US-DOI
Kurokawa et al. 2005 [13]	Prospective study	28	18:10	59.4	UICC	Tongue	T1 (*n* = 11) T2 (*n* = 12) T3 (*n* = 3) T4 (*n* = 2) N0 (*n* = 20) N1 (*n* = 6) N2a (*n* = 1) N2b (*n* = 1)	Preoperative	Echo Camera SSD-1200CV; Aloka, Tokyo, Japan	7.5 MHz	0.976
Songra et al. 2006 [14]	Prospective study	14	NR	NR	AJCC	Tongue/floor of the mouth	T1 N0 M0 (*n* = 8) T2 N2a M0 (*n* = 2) T4 N3 M0 (*n* = 1) T2 N0 M0 (*n* = 2) T2 N1 M0 (*n* = 1)	Intraoperative	HDI 5000; Advanced Technologies Ltd., Seattle	5–10 MHz	0.648
Mark Taylor et al. 2010 [15]	Prospective study	21	12:9	65	AJCC/UICC	Tongue/floor of the mouth	T1 (*n* = 5) T2 (*n* = 6) T3 (*n* = 6) T4 (*n* = 4)	Preoperative	NR	10–12 MHz	0.981
Iida et al. 2018 [16]	Retrospective study	56	34:22	59	NR	Tongue	NR	Preoperative	Model UST-5713T/Intraoperative Electronic Linear Probe; Hitachi Aloka Medical, Ltd., Tokyo, Japan	16 MHz	0.867
Noorlag et al. 2020 [17]	Retrospective study	146	74:72	64	AJCC/UICC	Tongue	T1 (*n* = 84) T2 (*n* = 62)	Preoperative	EpiQ 5 with CL15–7 transducer; Philips Medical Systems, Best, The Netherlands	15	0.78
Yoon et al. 2020 [18]	Prospective study	20	13:07	60.35	NR	Tongue	NR	Intraoperative	L15-7io Philips Healthcare; Philips North America Corporation, Andover, MA, USA	7–15 MHz	0.95
Bulbul et al. 2021[19]	Prospective study	23	15:8	59.1 ± 17.2	AJCC/UICC	Tongue	T1 (*n* = 13) T2 (*n* = 8) T3 (*n* = 2)	Intraoperative	L15-7io Philips Healthcare; Philips North America Corporation, Andover, MA, USA	7–15 MHz	0.9449
Filauro et al. 2021 [20]	Retrospective study	49	27 22	65.6 ± 15.8	AJCC/UICC	Oral cavity (buccal mucosa, tongue, floor of the mouth)	T1 (*n* = 15) T2 (*n* = 21) T3 (*n* = 13)	Preoperative	L15-7io Philips Healthcare; Philips North America Corporation, Andover, MA, USA	7–15 MHz	0.76
Harada et al. 2021 [21]	Retrospective study	128	85:43	55.7	NR	Tongue	NR	Preoperative	HI VISION Avius, Hitachi Healthcare Systems, Japan	13	0.815
Izzetti et al. 2021 [22]	Retrospective study	10	4 6	68.7 ± 10.2	AJCC/UICC	Oral cavity	Tis (*n* = 2) T1 (*n* = 3) T2 (*n* = 4)	Preoperative	Vevo MD; VisualSonics, Toronto, ON, Canada	70	0.96
Rocchetti et al. 2021 [23]	Retrospective study	36	23:13	62.0 ± 16.1 (M) 71.2 ± 10.6 (F)	AJCC/UICC	Oral cavity	Tis (*n* = 3) T1 (*n* = 9) T2 (*n* = 20)	Preoperative	E-CUBE 15 EX US scanner; Alpinion Medical Systems, Seoul, Republic of Korea	8–17 MHz	0.907
Caprioli et al. 2022 [24]	Retrospective study	41	25:16	64.07 ± 17.67	AJCC/UICC	Tongue	T1s (*n* = 5) T1 (*n* = 21) T2 (*n* = 15)	Preoperative	NR	7–22 MHz	0.84
Nilsson et al. 2022 [25]	Prospective study	40	25:15	65 ± 14	AJCC/UICC	Tongue	T1 (*n* = 19) T2 (*n* = 10) T3 (*n* = 11)	Preoperative	8870 probe, BK Medical Flex Focus 500 US; Peabody, MA, USA	18 MHz	0.6
Takamura et al. 2022 [26]	Retrospective study	48	28:20	65.7	AJCC/UICC	Tongue	T1 (*n* = 28) T2 (*n* = 20)	Preoperative	EUP-O54J transducer, HI VISION Preirus; Hitachi, Tokyo, Japan	7–13 MHz	0.83
Au et al. 2023 [27]	Retrospective study	19	NR	NR	NR	Tongue	NR	Preoperative	NR	NR	0.910
Kumar et al. 2023 [28]	Prospective study	50	37: 13	47.3 ± 11.7	AJCC/UICC	Tongue	T1 9 T2 32 T3 9	Preoperative	Aixplorer US system; SuperSonic Imagine, Aix-en-Provence, France	6–13 MHz	0.880

AJCC/UICC: American Joint Committee on Cancer/Union for International Cancer Control; NR: not reported; US: ultrasound.

## Data Availability

No new data generated.

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
