# Peer review of "The Reliability of Ultrasonographic Assessment of Depth of Invasion: A Systematic Review with Meta-Analysis"

_diagnostics, 2023, doi:10.3390/diagnostics13172833_

Round 1

Reviewer 1 Report

Dear Authors, a very important topic of intraoperative ultrasound assessment of DOI of OSCC is presented. Artifacts hinder in many cases adequate evaluation of DOI in CT and MRI preoperatively. The design of the study is clearly outlined and the structure well organized. 

Some minor comments: 

1. Figure 1 is not mentioned in the text. Content of Figure 1 should be explained in one figure legend. 

2. Discussion: DOI is already included in the UICC staging manual. Please clarify line 307-309. Line 310-316 is confusing to the reader as it is not relevant to DOI. Especially citation 32 does not add any relevant information. 

3. It would be interesting which transducers and/or frequencys and time point for intraoral US in OSCC is recommended. Maybe some inforamtion on this could be added in the discussion?

Thank you. 

Author Response

We thank the Reviewer for the comments, which helped to improve our manuscript. Please find below point by point response.

Dear Authors, a very important topic of intraoperative ultrasound assessment of DOI of OSCC is presented. Artifacts hinder in many cases adequate evaluation of DOI in CT and MRI preoperatively. The design of the study is clearly outlined and the structure well organized. 

We thank the reviewer for this positive remark on our work

Some minor comments: 

  1. Figure 1 is not mentioned in the text. Content of Figure 1 should be explained in one figure legend. 

We mentioned figure 1 at line 141 and provided figure legend at line 142.

  1. Discussion: DOI is already included in the UICC staging manual. Please clarify line 307-309. Line 310-316 is confusing to the reader as it is not relevant to DOI. Especially citation 32 does not add any relevant information. 

The sentence was edited as it follows: “From this perspective, DOI is an independent predictor of nodal metastases, and its inclusion in the staging system improved the decision making of elective neck dissection, especially in patients with early OSCC.”

  1. It would be interesting which transducers and/or frequencys and time point for intraoral US in OSCC is recommended. Maybe some inforamtion on this could be added in the discussion?

We thank the reviewer for this comment. Unfortunately, the available literature encompasses a wide variety of ultrasound frequencies for DOI evaluation, which was also reported as a limitation to our review. The frequency range of the included literature varies between 5 and 70 MHz, therefore at present we prefer not to recommend any frequency in particular. We edited the limitations paragraph as it follows in order to further clarify these aspects:

“The present study has some limitations. First, the variability of frequencies employed and ultrasound acquisition protocols may hinder comparison between studies, and thus the drawing of firm conclusions. However, it could be observed that current literature consistently reports the use of linear probes, although the variability in the frequencies employed may hinder the recommendation of a specific ultrasound frequency. Importantly, although some differences were detected in terms of and timing of ultrasonographic scan, it could be hypothesized that preoperative ultrasonography may prove beneficial for surgical planning. Nevertheless, the evidence on the intraoperative ultrasound acquisition supports a role of this technique in tumor resection with clear margins”

Reviewer 2 Report

Are there any standardized criteria for the measurement of DOI by ultrasonography?If there is one ,how did you deal with the difference of the measurements of DOI under circumstances of scanners of different frequencies?

Author Response

We thank the reviewer for the comments which helped to improve our manuscript. please find below point-by-point response.

Are there any standardized criteria for the measurement of DOI by ultrasonography?If there is one ,how did you deal with the difference of the measurements of DOI under circumstances of scanners of different frequencies?

We thank the reviewer for this comment. DOI by definition is the distance between the normal mucosal surface and the deepest margin of a neoplastic lesion in the tissue. For study selection, the definition for DOI was cautiously screened in order to discriminate studies describing other parameters, such as tumor thickness. However, no further assumptions can be made regarding the measurement of DOI depending on the ultrasound frequencies employed. the limitations paragraph was edited as it follows in order to clarify this aspect:

“although the definition for DOI was cautiously screened in order to discriminate studies reporting on other parameters (e.g. tumor thickness), great variability is encountered in the literature, hindering further assumptions regarding the measurement of DOI depending on the ultrasound frequencies employed.”